# The role of high-risk geographies in the perpetuation of the HIV epidemic in rural South Africa: A spatial molecular epidemiology study

**Diego F. Cuadros**[1,2]*, **Tulio de Oliveira**[3,4], **Tiago Gräf**[3,5], **Dennis M. Junqueira**[3,4], **Eduan Wilkinson**[3,4], **Philippe Lemey**[6], **Till Bärnighausen**[7,8,9], **Hae-Young Kim**[7,10], **Frank Tanser**[7,11,12,13]

**1** Department of Geography and Geographic Information Science, University of Cincinnati, Cincinnati, OH, United States of America, **2** Health Geography and Disease Modeling Laboratory, University of Cincinnati, Cincinnati, OH, United States of America, **3** KwaZulu-Natal Research Innovation and Sequencing Platform (KRISP), Nelson R Mandela School of Medicine, University of KwaZulu-Natal, Durban, South Africa, **4** School of Laboratory Medicine and Medical Science, Department of Health Sciences, University of KwaZulu-Natal, Durban, South Africa, **5** Fundação Oswaldo Cruz (FIOCRUZ), Instituto Gonçalo Moniz, Salvador, Brazil, **6** Department of Microbiology and Immunology, Rega Institute for Medical Research, University of Leuven, Leuven, Belgium, **7** Africa Health Research Institute, University of KwaZulu-Natal, Durban, South Africa, **8** Heidelberg Institute for Public Health, University of Heidelberg, Heidelberg, Germany, **9** Department of Global Health and Population, Harvard T.H. Chan School of Public Health, Boston, MA, United States of America, **10** Department of Population Health, New York University Grossman School of Medicine, New York, NY, United States of America, **11** School of Nursing and Public Health, University of KwaZulu-Natal, Durban, South Africa, **12** Lincoln International Institute for Rural Health, University of Lincoln, Lincoln, United Kingdom, **13** Centre for the AIDS Programme of Research in South Africa (CAPRISA), University of KwaZulu-Natal, Durban, South Africa

* diego.cuadros@uc.edu

**Data Availability Statement:** We use data collected by the Africa Health Research Institute (AHRI) Population Intervention Platform

## Abstract

In this study, we hypothesize that HIV geographical clusters (geospatial areas with significantly higher numbers of HIV positive individuals) can behave as the highly connected nodes in the transmission network. Using data come from one of the most comprehensive demographic surveillance systems in Africa, we found that more than 70% of the HIV transmission links identified were directly connected to an HIV geographical cluster located in a peri-urban area. Moreover, we identified a single central large community of highly connected nodes located within the HIV cluster. This module was composed by nodes highly connected among them, forming a central structure of the network that was also connected with the small sparser modules located outside of the HIV geographical cluster. Our study supports the evidence of the high level of connectivity between HIV geographical high-risk populations and the entire community.

## Background

Human Immunodeficiency virus (HIV), like all sexually transmitted infections, is dispersed through a population via sexual networks [1]. In such networks, highly connected individuals

Surveillance Area (PIPSA). AHRI considers requests for access to the AHRI Data Repository through a formal application process (https://data.ahri.org/index.php/home). Investigators are required to outline use of the desired data and/or biological samples. Prior to releasing data, researchers interested in using the PIPSA data will be required to sign an agreement stipulating that under no circumstances may they share PIPSA data with other researchers. Additionally, they will be required to sign a confidentiality agreement specifying that they will use the data only for their specified research purposes and they will not identify any individual participant. Investigators must also agree to use secure technology to safeguard the data. Data will be shipped directly to the investigators under the auspices of the AHRI. The data are supplied to investigators de-identified. For further information contact Dickman Gareta (dickman.gareta@ahri.org), the Research Data Management leader at AHRI.

**Funding:** This work was supported by two National Institute of Health (NIH) grants (R01HD084233, R01AI124389 and R21TW011687-01; DFC, FT, HY-K). The Africa Health Research Institute's Demographic Surveillance Information System and Population Intervention Programme is funded by the Wellcome Trust (201433/Z/16/Z; URL: https://wellcome.org/), and the South Africa Population Research Infrastructure Network (funded by the South African Department of Science and Technology and hosted by the South African Medical Research Council; URL: https://www.dst.gov.za/). The funders had no role in study design, data collection and analysis, decision to publish, or preparation of the manuscript.

**Competing interests:** The authors declare no competing interests.

are the key component for efficient transmission of the infection to new hosts [2, 3]. In most countries, HIV epidemics are concentrated in subpopulations (core groups) composed by these individuals with certain behavioral characteristics that put them at high-risk for the acquisition and onward transmission of the infection (e.g. female sex workers, injecting drug users, men who have sex with men among others) [4]. Therefore, control interventions are commonly designed to target these core groups to disrupt the viral transmission in the entire network [4, 5]. The epidemiology of HIV is marked by heterogeneous dynamics of spread across the globe. Compared to the rest of the world, the epidemic in Sub-Saharan Africa (SSA) has escaped from the high-risk sub-populations and spread into the general population [6]. SSA has by far the largest HIV epidemic in the world, with nearly one in every 20 adults (5%) living with HIV, accounting for roughly 70% of the global HIV disease burden [7]. The complex epidemiological context in this part of the world has prevented a clear identification of the cascade of social, behavioral and structural causes that have led to the spread of HIV across the general population in SSA [8].

Recent work has uncovered the extraordinary geographical heterogeneity of the risk of HIV in generalized epidemic settings [9, 10]. These findings identify priority geographic areas in which vulnerable populations at high risk are located and thus presents a real opportunity to apply spatially targeted interventions to most at risk population to maximize the impact on the epidemic in SSA [11]. In the last few years, a radical shift in thinking about geographical targeted interventions has prompted international agencies such as the President's Emergency Plan for AIDS Relief (PEPFAR) to include geographical prioritization as a key component of their overall HIV intervention strategy (PEPFAR 3.0) [11, 12].

However, some critical unexplored issues remain to be addressed in order to quantify the likely impact of geographically targeted interventions on the trajectory of the overall epidemic. The spatial connectivity of the transmission network of a rural community in SSA has never been studied before, and the contribution of geographical key-populations to the spread of the infection in the entire community is not well understood. We hypothesize that these high-risk locations behave as a core group that harbor highly connected nodes in the transmission network, and interventions targeting these geographical key-populations could not only reduce the levels of new infections in these geographical core groups, but also substantially disrupt the transmission of the infection in the whole community. If the level of connectivity between the geographical core groups and the individuals in the community is high, then a successful intervention approach in these high-risk locations could generate a substantial impact on reversing the overall epidemic in the entire community (S1 Fig in S1 Text).

We asked the question: what is the contribution of HIV high-risk locations in the overall HIV transmission network? Clear evidence of high degree of connectivity would provide real impetus for a distinct way of structuring the HIV targeted response in SSA settings. Strategies focusing resources to disrupt the HIV transmission network in such geographical key-populations could impact the whole population. To answer this critical question, we applied spatial epidemiology as well as phylodynamic methods to reconstruct a spatially-explicit transmission network of one of the Africa's largest population-based prospective cohorts located in the rural KwaZulu-Natal province of South Africa, the Africa Health Research Institute (AHRI) Population Intervention Platform Surveillance Area (PIPSA) [13].

## Methods

### Data sources

Data come from one of the most comprehensive demographic surveillance systems in Africa— PIPSA [13]. This surveillance system is located in Hlabisa, one of the five subdistricts in the

rural district of Umkhanyakude in northern KwaZulu-Natal, South Africa (S2 Fig in S1 Text). This surveillance system has routinely collected socio-demographic, behavioral and epidemiological information on a population of ~90,000 participants within a circumscribed geographic area (438 km2) for over a decade. Additional data were collected from population-based HIV surveillance and a sexual behavior survey that are conducted annually. All individuals ≥15 years of age are eligible to participate in this survey. Of those participants contacted, 60% agreed to be tested at least once. After obtaining written informed consent, for HIV testing field workers collected finger-prick blood samples and prepared dried blood spots according to the UNAIDS and WHO *Guidelines for Using HIV Testing Technologies in Surveillance* [14]. For each individual, the result of the most recent HIV test was assigned as the final HIV-serostatus of each participant. Participants included in the surveillance were geo-located to their respective homesteads of residence using the comprehensive geographic information system. To protect the confidentiality of the participants a geographical random error was introduced to the geographical coordinates of each homestead included in the study [15]. We created two different datasets to generate the geographical distribution of two epidemiological measures, HIV prevalence and incidence. To generate the dataset for HIV prevalence estimation, we utilized data sampling derived from information collected for phylogenetic analyses from 2011 to 2014. These four continuous population-based HIV surveillance datasets (2011–14) were combined to generate a robust dataset for further epidemiological and geographical analyses. As a result, the final sample included a total of 18,295 individuals (11,345 females and 6,945 males), from which 5,624 (4,279 females and 1,345 males) tested positive for HIV (S3 Fig in S1 Text).

To generate the dataset for HIV incidence estimation, we used a previous cohort generated to estimate the number of HIV sero-conversions from 2004 to 2014 in the Africa Centre's area of study [16]. This HIV cohort was composed by individuals ≥ 15 years of age who were observed to be HIV negative at a particular time-point during a routinely population-based HIV survey and agreed to be tested on at least one subsequent occasion. As a result, 17,984 individuals were included in the analysis [16]. Participants seldom test every year, and the median interval of time between last HIV-negative and first HIV-positive test is 2.18 years. The date of HIV seroconversion was assumed to occur midway between the date of the last negative and first positive HIV test. A total of 2,311 seroconversions were observed during the time of the population-based HIV surveys included in the study. Further description of the data sources can be found in Supplementary Materials.

## Sequencing and phylogenetic analyses

Under the assumption that transmissibility is maximized by high HIV viral load, and the amplification and sequencing of dried blood spots is mostly successful at high viral load, sequencing was prioritized for samples with >10,000 RNA copies/ml. In this study only HIV-1 sequences assigned as subtype C were included in the downstream phylogenetic analyses. In total, 1,426 individuals (25.4% of the total number of HIV-positive individuals) had HIV-1 *Pol* sequences that met internal quality control standards were classified as subtype C. Linkages of viral transmission among these individuals were estimated by phylogenetic analyses where sequences grouped with high support and showing low genetic divergence were considered as being isolated from individuals linked by transmission events, hereafter called phylogenetic transmission clusters. To account for possible linkages with individuals living outside the study area, the 1,426 HIV subtype C *Pol* sequences from the sampled area were analyzed in conjunction with a previously compiled [17] dataset of 11,216 background sequences isolated in southern African countries. Sequence alignment was performed with MAFFT [18] and a ML phylogenetic tree was inferred in FastTree2 [19] using the general time reversible (GTR)

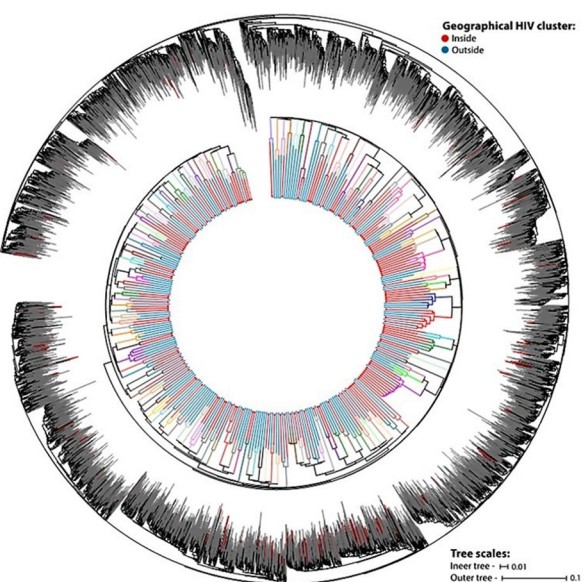

**Fig 1. HIV-1 *Pol* gene maximum likelihood trees.** The outer tree shows the identification of the target phylogenetic transmission cluster (red branches) in a background of a control dataset with sequences from Southern African countries (gray branches). The inner tree depicts the identified clusters (in different colors) and the geographical location of the sampled individuals (according to the legend at the top right).

substitution model with gamma-distributed rate variation among sites. Branch support was evaluated using Shimodaira-Hasegawa test (SH-like) with 1000 bootstrap replicates. For visualization and tree annotation the web application iTOL [20] was used. From the phylogenetic tree, HIV transmission clusters were identified with Cluster Picker software [21], by selecting clades with branch support higher than 90 (SH-like) and within-cluster pairwise genetic distance smaller than 4.5%. Aiming to investigate the HIV-1 transmission dynamics within Africa Centre study area, here we only analyzed phylogenetic clusters the met the above-mentioned criteria and were composed by more than 70% of sequences sampled from individuals living in the area. Therefore, the phylogenetic transmission clusters analyzed here represent mostly the ARHI area internal circulation of viral strains. From the 1,426 HIV *Pol* sequences included in the phylogenetic analyses, the homestead geo-location of the sampled individual was available for 1,222 sequences (Fig 1). Further description of the phylogenetic analyses can be found in Supplementary Materials.

## Identification of the high-risk communities

We generated continuous surface maps of HIV prevalence and HIV seroconversion in the area of study using a moving two-dimensional Gaussian kernel of 3 km search radius to produce robust epidemiological estimates that vary across continuous geographical space to generate a grid of 300m x 300m pixels. The size of the kernel was determined from the results of previous work [15]. The kernel moves systematically across the map and measures spatial variation in HIV prevalence (or incidence) across the surveillance area.

We identified the HIV geographical high-risk location (geospatial areas with significantly higher numbers of HIV positive individuals) in this rural community through Kulldorff spatial scan statistics analysis [22] for clustering detection using the 2011–14 population-based HIV surveillance dataset. This methodology has been widely applied in public health research [9, 23–26], and a more technical description behind the spatial scan statistics have been discussed

elsewhere [25]. Briefly, this technique is based on a cluster detection test designed to identify areas with higher numbers of cases (i.e. HIV-positive individuals) than would be expected under the assumption of random distribution of the cases in space, and then evaluate their statistical significance by gradually scanning a circular window that moves across the entire study region. The detection of geographical HIV clusters with Kulldorff spatial scan statistics analysis is achieved by testing each potential cluster against the null hypothesis that the distribution of cases was proportional to the population size using likelihood ratios and t-tests. Therefore, one of the most important features of Kulldorff spatial scan statistics analysis is that it accounts for the underlying population at risk to control for the uneven population distribution in the area of study. The radius of the searching circle is continuously changing, and it can take any value from 0 up to a pre-specified maximum value. For our study, the maximum default value of < 50% of the total population included in the window was used. The statistical significance of each potential cluster was then estimated using a likelihood ratio test. This test is computed assuming that the number of HIV-positive individuals in each circular window is an independent Bernoulli random variable. A null hypothesis of spatial randomness is then used to compare the numbers of observed and expected HIV-positive individuals within and outside the circular window. Circular windows with the highest likelihood ratio values were identified as potential clusters. Geographical HIV clusters (geospatial areas with significantly higher numbers of HIV positive individuals) with a $P < 0.05$, calculated through Monte Carlo simulations, were identified as statistically significant, and were further analyzed for additional epidemiological description.

## Network analysis

We assessed the number of transmission links that arose from within the HIV geographical high-risk location identified with Kulldorff spatial scan statistics analysis. For privacy protection and ethical considerations, we generated maps without the inclusion of any geographical reference. Likewise, spatial random error in the geographical references of all linkages of viral transmission identified was introduced, and the maps are shown for illustrative purposes only. With the same aim of privacy protection and with the purpose of performing network analyses, we generated a grid composed by 108 cells of 3km X 3km dimension that covered the entire surveillance area (S2C Fig in S1 Text). We used the grid to aggregate the location of the transmission links into the centroids of each of the 108 cells that represented the nodes of the network. We conducted several network analyses to identify the influence of the HIV high-risk location in the network configuration arisen from the transmission links identified. First, we assessed the association between the HIV prevalence estimated for each cell of the grid and the node degree of the corresponding cell using simple Pearson correlation analysis and bivariate maps. Second, we assessed the eigencentrality of the nodes to identify the influence of each node in the network based on the number of links it had to other nodes in the network (node degree), and also how many links their connections had through the network. In this analysis, nodes with high eigenvector centralities are those connected to other nodes that in turn are linked to highly connected nodes, identifying high-density substructures. We also conducted a community structure detection analysis to identify modules (communities) of nodes densely connected to themselves but sparsely connected to other modules.

## Microsimulation models

To assess the randomness of the spatial configuration of the transmission links observed we ran two microsimulation models with different parameters of spatial HIV transmission link formation to best explain the transmission patterns observed. Both microsimulations used

data from the total HIV-positive population, in which the geo-location of each participant as well as the number of transmission links were used. We generated random model, in which the probability of a link formation between individuals was independent of the distance between individuals or the location of the individuals related to the geographical HIV cluster. We explored the influence of the geographical HIV cluster in the network configuration by generating a gravity model in which individuals are selected based on their location with respect to the geographical HIV cluster, and individuals located within the geographical HIV cluster had higher likelihood of being selected for a link formation. Then, the distance between individuals is measured, and the location of the individuals related to the geographical HIV cluster (if the individuals are located within or outside the geographical HIV cluster) is recorded. In each microsimulation, we recorded the number of links subject to the location of participants with respect to the HIV high-risk location (within, outside, or in both sides of the HIV cluster). All results were based on the average of 10,000 realizations of the model to estimate the median and the 95% credible interval (CrI). Further description of the microsimulations can be found in Supplementary Materials. Network analyses and microsimulations were conducted using R and MATLAB, and spatial maps were generated using ArcGIS 10.5.

### Ethic statement

Members of households resident within the program area are eligible for the study enrolment. ACDIS staff recruit participants based on their residential address using standard operating procedures for the AHRI's on-going population-based study. All individuals ≥15 years of age who are resident members of households in the program area are eligible to be included. Of those participants contacted, a written informed consent is sought for all individuals before enrolment. Consent process may involve structured verbal interactions, brief leaflets, or use of story boards or videos to actively engage with participants and ensure their understanding. The study team developed six short video clips, delivered in Zulu language, to help improve and standardize informed consent process. There are no restrictions on race/ethnicity, social background, or gender in the inclusion of individuals participating in this surveillance system. All study participants must provide informed consent prior to enrollment and participation in the surveillance. Data collected for the ACDIS database are used for research purposes only. All ACDIS data collection is administered and supervised by trained personnel in a controlled environment. A longitudinal population-based platform for epidemiology and intervention research (BE290/16) was approved by the Biomedical Research Ethics Administration at the University of Kwazulu Natal, South Africa.

## Results

### Sequencing and phylogenetic analyses

From the 1,426 HIV Pol sequences included in the phylogenetic analyzes collected from 2011 to 2014, the homestead geo-location of the sampled individual was available for 1,222 sequences. From these 1,222 individuals we identified that 333 were linked in 132 phylogenetic transmission clusters (Fig 1) with sizes ranging from 2 sequences to 11 (S7A Fig in S1 Text). These clustered individuals accounted for a total of 350 transmission links, whose geo-location is illustrated in S8A Fig in S1 Text. The geo-location of non-phylogenetically linked individuals is illustrated in S8B Fig in S1 Text. The proportion of individuals sampled from inside the identified HIV high-risk location were roughly the same among individuals in phylogenetic clusters and in the whole dataset of sequences (S7B Fig in S1 Text).

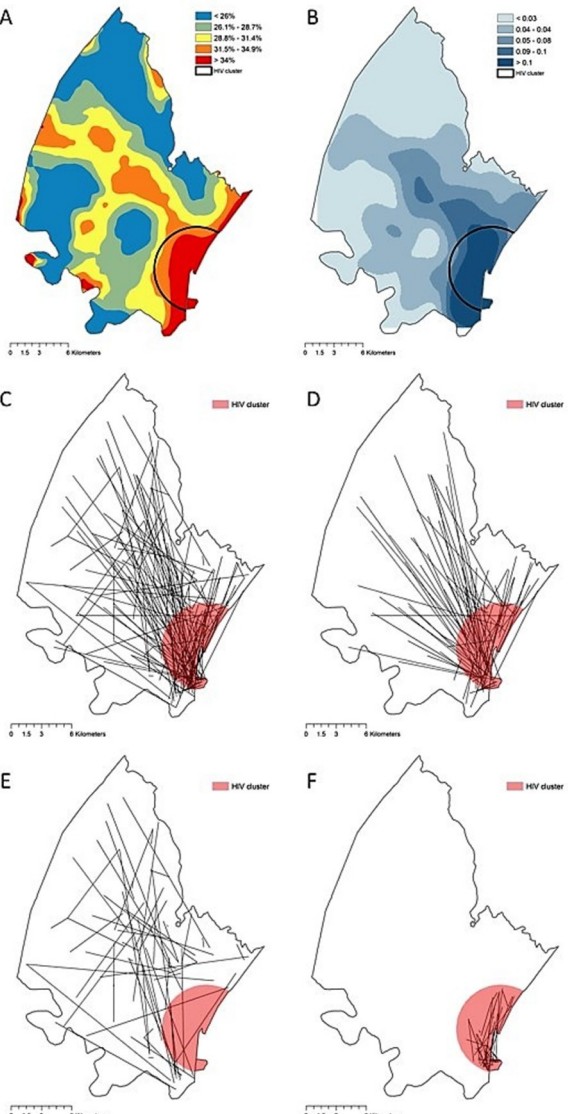

**Fig 2. A)** Kernel interpolation of the HIV prevalence in the area of study. **B)** Estimated HIV sero-conversion per pixel (300m x 300m). **C)** All transmission links. **D)** Transmission links in which at least one individual is located inside the HIV high-risk area. **E)** Transmission links outside of the HIV high-risk area. **F)** Transmission links within the HIV high-risk area. Red circle encloses the geographical HIV high-risk area (HIV geographical cluster). Spatial random error in the geographical references of all linkages of viral transmission identified was introduced, and thus do not represent the actual spatial location of the links, and maps were generated for illustrative purposes only. Maps were created using ArcGIS by Esri version 10.5 (http://www.esri.com) [39].

## Spatial analysis

The Kulldorff spatial scan statistics identified an HIV geographical cluster (high-risk location) containing a higher-than-expected proportion of HIV-infected individuals situated at the southeastern part of this rural community (Fig 2A). While this HIV geographical cluster contained 32% (5,854) of the total sampled population and bounded only 7.5% of the study area, it harbored 40.8% (2,295) of the total HIV-positive individuals. The HIV prevalence estimated within the HIV high-risk location was 39.1% (95% confidence interval [CI]: 37.9–40.4) compared to 26.8% (95% CI: 26.0–27.6) estimated outside of this HIV high-risk area. Likewise,

estimated HIV sero-conversions was highly concentrated within the HIV high-risk location (Fig 2B), with an average of 0.175 seroconversions per pixel (300 m x 300 m), compared to 0.035 seroconversions per pixel in the area outside of the HIV high-risk location. Likewise, the HIV incidence estimated within the HIV high-risk location was 4.33 infections per 100 person-years, compared to 2.85 infections per 100 person-years estimated outside of this area.

## Network analysis

Of the 350 phylogenetically paired transmission links identified (Fig 2C), 254 links (72.6%) included at least one individual located within the HIV high-risk location identified by SaTS-can (Fig 2D). The average distance between individuals genetically linked was 6.4 km, and only six transmission links (1.7%) were identified between individuals in the same household (S6 Fig in S1 Text). We also identified 29 heterosexual HIV-seroconcordant couples living in the same household but not genetically linked (S6 Fig in S1 Text). Network configuration resulted from the spatial aggregation using the grid is illustrated in S2C Fig in S1 Text, and nodes were labeled using sequential numbers. Node degree ranged from one (nodes with no connections were excluded from analysis) to 44 (node 15). The average degree of the nodes located within the HIV high-risk location was 20, ranging from nine links from node 33 to 44 links from node 15, whereas the average degree of nodes located outside the HIV high-risk area was five, ranging from one in several nodes to 18 in node 49.

Likewise, we found a statistically significant positive correlation between the average HIV prevalence of the cell and the node degree (Pearson correlation coefficient = 0.69; $p < 0.005$; Fig 3A). Nodes 6, 15, 16, 24, 25 and 34, were the nodes located within the HIV-high risk location with the highest HIV prevalence ranging from 34.6% in node 34 to 47.7% in node 16 (Fig 3B). Likewise, these nodes were highly connected nodes, with node degree ranging from 17 links in node 16 to 44 links in node 15 (Fig 4A). Eigencentrality analysis showed that node 15

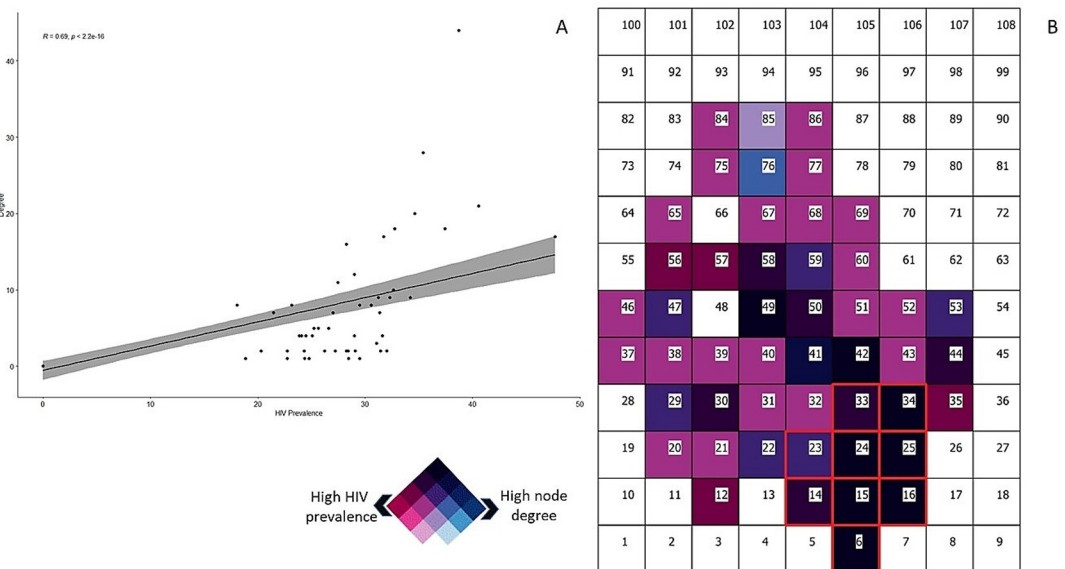

**Fig 3. Association between node degree and HIV prevalence of the corresponding cell.** A) Correlation between HIV prevalence of the node and the corresponding node degree. The line illustrates the regression line showing the positive association between the node connectivity (degree) and the HIV prevalence of the corresponding cell (p<0.005). B) Bivariate map showing areas with high node degree and high HIV prevalence (dark cells). Numbers within the cells illustrate the labels of each cell (node). The nodes included within the HIV high-risk area are delineated in red.

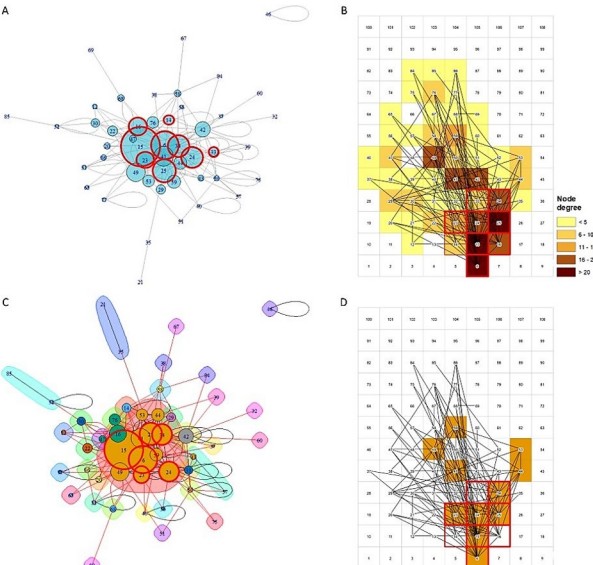

**Fig 4. Spatial network configuration of stable sexual partnerships.** A) Eigenvalue estimations for each node. Nodes with highest eigenvalues are centered close to the nodes with more links. Circles delineated in red indicate the nodes located within the HIV cluster. B) Map illustrating the node degree of each pixel. Nodes with high degree are illustrated in dark brown, whereas nodes with low degree are illustrated in yellow. C) Illustrates the different node communities identified in the network. Circles delineated by red illustrate the nodes located within the geographical HIV high-risk area. D) Map illustrating the location of the cells included in the main node community of the network (brown cells). The numbers in all figures represent the labels of each node in the network.

had the highest eigenvector centrality (1.00) followed by node 6 (0.70), node 25 (0,58), node 34 (0.55), and node 24 (0.53), all of them located within the HIV high-risk area (Fig 4B). Community detection analysis identified one single large central module formed by nodes 6, 15, 23, 24, 25, 34, 41, 44, 49, 53, and 59 (Fig 4C). Six of these 11 nodes were located within the HIV high-risk location (Fig 4D). The other nodes in the network did not form more communities and were sparsely connected, and most nodes located outside of the HIV high-risk area were and only connected through the large central module identified within this HIV cluster.

## Microsimulation models

Results from the microsimulations indicated that the observed HIV transmission link configuration did not follow a random pattern (Fig 5A–5C). The random model overestimated the percentage of transmission links occurring outside of the HIV high-risk location and underestimated the percentage of transmission links happening within this high-risk location. Conversely, the gravity model illustrated better the spatial configuration of the network observed (Fig 5D). This model estimated that individuals that were located within the HIV high-risk location had 20% higher probability of forming a transmission link compared to individuals located outside this area.

## Discussion

In this study we assessed the connectivity of the spatially-explicit transmission network of a community with a HIV hyper-endemic epidemic. We found that a HIV geographical cluster (geospatial areas with significantly higher numbers of HIV positive individuals) in a peri-urban area characterized by high HIV prevalence and incidence compared to other areas is

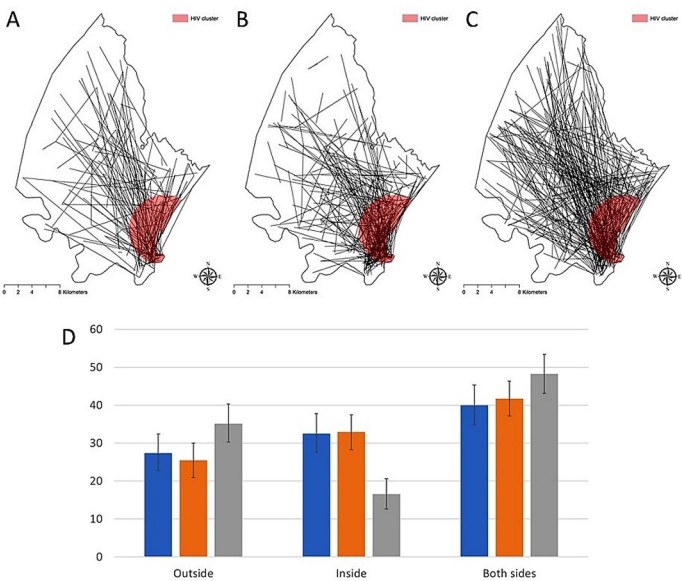

**Fig 5. Comparisons among link estimations derived from the microsimulation models. A)** Spatial distribution of the HIV transmission links from the data. **B)** Spatial distribution of the HIV transmission links using the gravity model **C)** spatial distribution of the HIV transmission links using the random model. **D)** Comparisons between the percentage of HIV transmission links from individuals located outside, individuals located inside, and links formed by one individual located inside and the other individual located outside of the HIV high-risk area. Blue bars illustrate the estimations from the data, orange bars illustrate estimations from the gravity model, and blue bars illustrate estimations from the random model. Spatial random error in the geographical references of all linkages of viral transmission identified was introduced, and thus does not represent the actual spatial location of the links, and maps were generated for illustrative purposes only. Maps were created using ArcGIS by Esri version 10.5 (http://www.esri.com) [39].

not only an area with high burden of the disease, but it is also a key component of the entire network configuration, i.e. the sexual network connectivity of the whole area. We found that more than 70% of the HIV transmission links identified by phylogenetic analysis were directly linked to the HIV high-risk location. Moreover, we identified a single central large community of highly connected nodes located within the HIV geographical cluster. This module was composed by nodes highly connected among them, forming a central structure of the network that was also connected with the small sparser modules located outside of the HIV geographical cluster. Whilst the linked infections identified do not represent direct transmissions in the vast majority of cases (could have gone through an unidentified intermediate individual), the viral linkages map the spatial configuration of the transmission network and are indicative of substantial mixing between the population living in the high-risk location and the entire community. Furthermore, we found a similar network pattern configuration using data from stable partnerships formation in the study area. For those who formed a stable sexual partnership within the surveillance area, we found that the spatial configuration of the sexual contact network was remarkably similar to the pattern observed from the HIV transmission network reconstructed in this study using phylogenetic methods, and about 76% of the links formed by stable sexual partnerships were directly linked to the HIV geographical cluster. Moreover, the configuration of the links generated by stable sexual partnerships also formed a similar large highly connected community of nodes was located within the HIV geographical cluster. Likewise, as our modeling results suggest, the high HIV burden area identified (HIV geographical cluster) is an area of high attractiveness for link formation, potentially driven by high urban development and other socio-economic factors that influence high human mobility to the

high-risk location (section 3.4 in Supplementary Materials). Collectively these results suggest that the main network module located within the HIV geographical cluster might behave as the highly connected hub of the network that enhances the connectivity of the entire network in this community.

The peri-urban area where the HIV geographical cluster was located was characterized by a high proximity to the main N2 highway, higher household wealth, and more young adults aged 25–34 years residing in this area. We also have previously reported that >50% of external in-migrants moving into the surveillance settled in the geographic HIV cluster [27]. In a previous study in which we longitudinally examined the spatial and temporal HIV incidence and prevalence patterns, HIV incidence in the communities outside the cluster have gradually declined by >70%, while the current HIV cluster persistently had one of the highest HIV incidences over the 15 years [27]. These reflect that the structural economic activities in the peri-urban area where the HIV cluster is located serve as a potential hub for the social and sexual network [28], especially for young adults, and expose them for a consistently high risk of HIV acquisition. A recent study which used the national Demographic and DHS data in seven Eastern and Southern African countries reported similar findings that environmental factors such as global human footprint and urbanization, representing the key constructs of economic activities, were the most important determinants and predictors of the spatial cluster of HIV prevalence beyond individual sociodemographic or sexual behaviours [29].

Previous similar studies conducted in fishing communities in Rakai, Uganda have assessed the cross-community infection arising from the high-prevalence fishing communities in Lake Victoria to the lower prevalence inland population [30–33]. Contrary to our findings, these studies found little connectivity between the fishing communities with high HIV prevalence and inland agrarian and trading communities with low HIV prevalence, and only less than 11% of the transmission links were identified between the high-prevalence and low-prevalence communities [31]. There are several differences between this study and our study. First, the main goal of the Rakai study was to assess the flow of infection from high HIV prevalence communities (fishing communities) to low prevalence communities (inland communities) at regional level. Therefore, data collected from 40 communities distributed in a large geographical region were included in the study. Conversely, our study focused on identifying the HIV transmission network configuration in a single well-defined local community located in a small well characterized peri-urban area. Second, locations where the fishing communities reside were identified as the high HIV burden areas, whereas in our study we implemented a well-known spatial statistical method to delineate the geographical area containing the highest burden of the infection in the local community residing in the peri-urban area. Lastly, since communities in the Rakai study were sparsely distributed across a large geographical region, these communities were mostly connected by migrant individuals [30], and thus the study focused more in depicting the migration network among these communities rather than the contact network among individuals. In contrast, our study illustrated the HIV transmission network potentially emerging from the configuration of the contact sexual network in this local community in rural Kwazulu-Natal.

Our results suggest that populations at high risk of HIV infection located within the HIV geographical clusters characterized by high HIV prevalence and incidence might be key components of the network configuration that sustain the dispersion of the disease in the entire network in some hyperendemic settings. Therefore, perturbations introduced in the network can propagate more effectively and reach more elements of the network if they affect these highly connected nodes located within this HIV high-risk area. If the level of connectivity between the HIV high-risk location and the entire community is high as it was shown in our study, then a successful intervention approach in these geographical key populations could

generate a marked impact on reversing the overall epidemic. In the HIV response of a country, the results presented in our study provide valuable information in support of a complex decision process that may also include, among several other elements, budgetary constraints along with political priorities beyond health outcomes. UNAIDS aim to ensure that "no one is left behind" [34] and this applies to individuals in these communities that are currently not being reached by the wider prevention response. Equally, these results provide critical information for the empowerment of individuals living in these communities to understand their risk and to take active steps to protect not only themselves but also to ultimately impact on the overall risk in the wider population. Disrupting the HIV transmission network using geographically targeted interventions to empower key communities could become a highly effective strategy aimed to optimize resources and maximize the impact in the fight against HIV in SSA. However, although the social implications of geographical community microtargeting has not been well stablished, it is important to highlight those social measures might need to be taken into consideration to avoid stigma and discrimination among these local communities. Therefore, further studies assessing the social impact of geographical targeted interventions are guaranteed.

## Limitations of the study

Several limitations of our study are worth noting. The relationship between phylogenetic tree topologies and the structure of the underlying contact network on which those topologies are generated is an ongoing and critical area in HIV phylogenetic research [35]. Interpretation of phylogenetic cluster distribution as indicators of contact network phenomena is complicated by often biased and/or incomplete sampling of transmission networks. HIV transmission networks are usually under sampled, and the proportion of sequences that phylogenetically cluster hardly exceeds 30% [36], as it was also observed in our study. Moreover, it is important to note that sampling can be biased by population engagement in care and previous studies have suggested that phylogenetic clusters using genetic distance thresholds tend to group individuals diagnosed early after infection [37, 38]. Although phylogenetic analyses can reveal paths in the contact network through which HIV spreads and provide a useful heuristic for assessing how well networks are connected within and between different subpopulations or risk-groups, phylogenetic analyses do not directly reveal the structure of transmission networks. In the absence of a fully sampled transmission network and detailed contact history, it is not possible to draw direct linkage between two individuals using phylogenetic analysis alone. Therefore, to overcome these methodological challenges, it would be important to integrate behavioral, epidemiologic and molecular sequencing data into a model that can identify the main parameters and processes intrinsic to the dynamics of the transmission in a contact network. Such models would facilitate a deeper understanding of HIV transmission dynamics in a spatially-explicit transmission network of an entire community. Combination of phylogenetic studies like the one presented here with behavioral studies reconstructing the contact network using partnership formation data like the study conducted by Kim and collaborators would depict a more accurate representation of the transmission network in which the HIV infection flows in a community suffering a hyperepidemic in rural South Africa.

Since date of infection could be inferred only for few individuals, we used the time difference between the date of the first HIV positive test and the date of the sampling that generated sequence data to test if individuals in a cluster are more engaged in care. This time difference has a median of 1.84 years (4.29 inter quartile range [IQR]) for clustered individuals and 2.24 years (5.64 IQR) for non-clustered individuals, representing a difference of around five months and suggesting a small impact of sampling bias due to engagement to care in the

analyses. Another potential limitation is the scarce availability of people mobility data in the area. This could be very useful to test our hypothesis that the geographical HIV cluster has a gravity effect and attracts both recipients and transmitters to its geographical region. For example, future studies should try to retrieve data from mobile phone tower usage and utilize this to model human mobility and daily migrations in the study area and assess how this could influence disease dynamics.

## Conclusions

In this study, we analyzed for the first time the spatially explicit HIV transmission network of a community with a generalized HIV epidemic in South Africa and the transmission intensity from the population at high-risk of infection. Through geospatial mapping technologies and the use of large-scale population-based surveys combined with phylogenetic analysis, we found that the key-populations at risk located in a peri-urban area suffering a high HIV burden are also highly connected populations that might play a significant role in the diffusion of the virus across the entire transmission network, which may help to enhance the equity of the HIV response. These insights highlight the potential benefit of microtargeting geographical units specific to these key populations at risk to maximize the public health impact with existing resources in the shortest possible time. Control interventions targeted directly to these geographical key populations might potentially propagate across the entire community located in areas with lower burden of the infection, a strategy that could benefit the overall community in the fight against HIV with the promise that no one is left behind.

## Supporting information

**S1 Text. Supporting methods and results.**
(PDF)

## Author Contributions

**Conceptualization:** Diego F. Cuadros, Tulio de Oliveira, Eduan Wilkinson, Till Bärnighausen, Hae-Young Kim, Frank Tanser.

**Data curation:** Diego F. Cuadros, Dennis M. Junqueira.

**Formal analysis:** Diego F. Cuadros, Tulio de Oliveira, Tiago Gräf, Dennis M. Junqueira, Eduan Wilkinson, Philippe Lemey, Hae-Young Kim.

**Funding acquisition:** Diego F. Cuadros, Tulio de Oliveira, Frank Tanser.

**Investigation:** Diego F. Cuadros, Tulio de Oliveira, Tiago Gräf, Dennis M. Junqueira, Philippe Lemey, Till Bärnighausen, Frank Tanser.

**Methodology:** Diego F. Cuadros, Tiago Gräf, Dennis M. Junqueira, Eduan Wilkinson.

**Supervision:** Till Bärnighausen, Frank Tanser.

**Validation:** Diego F. Cuadros.

**Visualization:** Diego F. Cuadros.

**Writing – original draft:** Diego F. Cuadros.

**Writing – review & editing:** Tulio de Oliveira, Tiago Gräf, Dennis M. Junqueira, Eduan Wilkinson, Philippe Lemey, Till Bärnighausen, Hae-Young Kim, Frank Tanser.

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
