## [Decision Letter · Decision Letter 0]

23 Sep 2021

PGPH-D-21-00177

The role of high-risk geographies in the perpetuation of the HIV epidemic in rural South Africa

Dear Dr. Cuadros,

Thank you for submitting your manuscript to PLOS Global Public Health. After careful consideration, we feel that it has merit but does not fully meet PLOS Global Public Health’s publication criteria as it currently stands. Therefore, we invite you to submit a revised version of the manuscript that addresses the points raised during the review process.

We look forward to receiving your revised manuscript.

Kind regards,

Sanghyuk Shin

Academic Editor

Journal Requirements:

1. We note that participants provided oral consent. Please state in the Methods:

- Why written consent could not be obtained

- Whether the Institutional Review Board (IRB) approved use of oral consent

- How oral consent was documented

For more information, please see our guidelines for human subjects research: https://journals.plos.org/plosone/s/submission-guidelines#loc-human-subjects-research

2. Please provide a detailed Financial Disclosure statement. This is published with the article, therefore should be completed in full sentences and contain the exact wording you wish to be published.

i). Please include all sources of funding (financial or material support) for your study. List the grants (with grant number) or organizations (with url) that supported your study, including funding received from your institution. 

ii). State the initials, alongside each funding source, of each author to receive each grant.

iii). State what role the funders took in the study. If the funders had no role in your study, please state: “The funders had no role in study design, data collection and analysis, decision to publish, or preparation of the manuscript.”

iv). If any authors received a salary from any of your funders, please state which authors and which funders.

3. Since your data is not available for proprietary reasons, please explain via email why the data is not available. Please also include the contact information for the third party organization that should be contacted should other researchers want to request access to this data and please include the full citation of where the data can be found. We also request that you verify with us via email that any researcher will be able to obtain the data set in the same manner that the you have obtained it. If you feel you are unwilling or unable to adhere to this policy, please explain your reasons by return email and your exemption request will be escalated to the editor for approval. Your exemption request will be handled independently and will not hold up the peer review process, but will need to be resolved should your manuscript be accepted for publication. One of the Editorial team will be in touch if they require more information.

4. Please provide separate figure files in .tif or .eps format only, and remove any figures embedded in your manuscript file. If you are using LaTeX, you do not need to remove embedded figures.

For more information about figure files please see our guidelines: https://journals.plos.org/globalpublichealth/s/figures

5. Please provide us with a direct link to the base layer of the map used in Figures 2, 5, Figures S1, S2, S7, S9, S10, S11, S16, S17 and ensure this location is also included in the figure legend. 

Please note that, because all PLOS articles are published under a CC BY license (creativecommons.org/licenses/by/4.0/), we cannot publish proprietary maps such as Google Maps, Mapquest or other copyrighted maps. If your map was obtained from a copyrighted source please amend the figure so that the base map used is from an openly available source.

Please note that only the following CC BY licences are compatible with PLOS licence: CC BY 4.0, CC BY 2.0  and CC BY 3.0, meanwhile such licences as CC BY-ND 3.0 and others are not compatible due to additional restrictions. If you are unsure whether you can use a map or not, please do reach out and we will be able to help you. 

The following websites are good examples of where you can source open access or public domain maps:

Additional Editor Comments (if provided):

I concur with the reviewers that this is a clearly written manuscript of a study using multiple robust methods to investigate an important public health problem. However, as one of the reviewers noted, the data availability statement does not appear to meet the requirements of the journal: https://journals.plos.org/globalpublichealth/s/data-availability. I realize that this study is based on highly sensitive HIV data. However, please note that a non-author institutional contact at AHRI should be specified in the manuscript: "When possible, we recommend authors deposit restricted data to a repository that allows for controlled data access. If this is not possible, directing data requests to a non-author institutional point of contact, such as a data access or ethics committee, helps guarantee long term stability and availability of data. Providing interested researchers with a durable point of contact ensures data will be accessible even if an author changes email addresses, institutions, or becomes unavailable to answer requests."

Reviewers' comments:

Reviewer's Responses to Questions

**Comments to the Author**

1. Does this manuscript meet PLOS Global Public Health’s publication criteria? Is the manuscript technically sound, and do the data support the conclusions? The manuscript must describe methodologically and ethically rigorous research with conclusions that are appropriately drawn based on the data presented.

Reviewer #1: Yes

Reviewer #2: Yes

2. Has the statistical analysis been performed appropriately and rigorously?

Reviewer #1: Yes

Reviewer #2: I don't know

3. Have the authors made all data underlying the findings in their manuscript fully available (please refer to the Data Availability Statement at the start of the manuscript PDF file)?

Reviewer #1: Yes

Reviewer #2: No

4. Is the manuscript presented in an intelligible fashion and written in standard English?

Reviewer #1: Yes

Reviewer #2: Yes

5. Review Comments to the Author

Reviewer #1: Comments:

Authors present analysis and identification of high-risk HIV geographical clusters and their role in HIV-1 transmission networks. They make recommendations that identifying high risk geographic spots could aid in intervention programming especially in Sub-Saharan Africa where the epidemic seems to be driven by geographical sources and sinks. In their attempt to answer the Questions “what is the contribution of HIV high-risk locations in the overall

HIV transmission network” they we applied spatial epidemiology and phylodynamic analysis to transmission network analysis one of the largest population-based prospective cohorts located in the rural KwaZulu-Natal province of South Africa. The authors demonstrated that spatial and transmission network analysis can identify specific geographical a HIV hyper-endemic epidemic. A large proportion the HIV transmission links identified by phylogenetic analysis were directly linked to the HIV high-risk location. This work is critical for targeting interventions for optimal use of scarce resources.

Comments

1. Although the authors clearly explain how the attempted to protect the confidentiality of the participants by introducing a geographical random error to the geographical coordinates of each homestead included in the study, the longitudinal cohort is well known and well established. What additional measures or community engagement were used to avoid social injury or stigma attached to the community if the results of this work can be translated back for intervention programming as suggested by the authors for “microtargeted” interventions?

2. What was the median between the date of the last negative and first positive HIV test for the incidence estimation cohort?

3. What is the proportion of participants with VL <10,1000 not sequenced? What is the potential impact for missing sequences due to lower viral loads (<10, 000 copies/mL) or sequencing prioritisation for higher viral loads on the identification of geographical clusters. Could authors provide analysis of possible impact of these values. Viral loads as low as 2000 copies/mL have associated with onward transmission.

4. Sequencing and phylogenetic analysis – what was the definition of a cluster used? What thresholds were used?

5. Figure 3A – not sure of the Pearson corr was the most appropriate given a number of outliers in the dataset.

Reviewer #2: Review

This paper blends spatial epidemiology and phylogenetic methods to investigate the spread of HIV in the Hlabisa region of South Africa. They demonstrate that a small region with the highest HIV prevalence and incidence also turns out to the location of many highly connected nodes in an Hiv phylogeny for the region. The methods were novel and interesting and most of the paper as written clearly; the results were interesting ad the figures are great. Please include line numbers in any future manuscripts!

Major comments

- Phylo/Network analysis. No description of what constitutes a cluster or how transmission links were assessed. The reader is pointed to the Sup Info but even in the Sup Info I could not find a clear explanation of either. It just says “the Phylotype approach” (with no reference). Please describe in more detail and clearly how transmission links were assessed, in the main body of the manuscript. I think you then also create a genetic network from the phylo links, but this is not explained either

- How do relative epidemic sizes influence these results? E.g. prevalence is highest in your geo cluster so you just have more cases from there, therefore more people link to it

Abstract

- Ambitious sounding and exciting, but in the first sentence I read “HIV geographical clusters” and I am not sure what that means. I think the terms in the abstract probably need to be clearly stated ( geo cluster, transmission links). In fact, although the authors are careful with their use of words throughout, they need to explicitly explain what they are calling a cluster because the term cluster already had a meaning in HIV phylogenetics.

Background

- No references to any published spatial epi/phylo papers, e.g. https://pubmed.ncbi.nlm.nih.gov/34159215/

- Typo in last line of background (or should be of)

Methods

- P6 para 3 than it would be expected should be than would be expected

- Add a subsection for microsimulation, even if most of that detail is in supplementary. Also include a brief explanation here of random vs gravity and what they mean because it’s important to understanding the results.

Results

- Figure 2 legend D) should be bolded, ad a full stop at the end but great figure. Maybe clarify in legend that “HIV cluster” means “HIV geographical cluster”

- P 10 top – you refer to figure Sup2D, but that figure doesn’t exist

Discussion

- First sentence problem with grammar

- P11 second paragraph, do you mean the geographic HIV cluster?

- The discussion is too long, slightly repetitive and the sentences are too long, I suggest you try to tighten it up a bit.

- The time from infection and directionality results come into the discussion but have not been discussed in the results only in the supplementary materials, and their relevance to the manuscript is unclear without reading the supplementary

References

- Ref 4 is missing

Supplementary Materials

- Amazingly detailed and clear, I really applaud your efforts to explain everything in so much detail

- Don’t really understand the order of the figures in relation to the main manuscript

- Very extensive – some things don’t seem to even be in the main manuscript, like directionality of links

6. PLOS authors have the option to publish the peer review history of their article (what does this mean?). If published, this will include your full peer review and any attached files.

**Do you want your identity to be public for this peer review?** For information about this choice, including consent withdrawal, please see our Privacy Policy.

Reviewer #1: No

Reviewer #2: No

---

## [Editor Report · Decision Letter 1]

5 Nov 2021

PGPH-D-21-00177R1

The role of high-risk geographies in the perpetuation of the HIV epidemic in rural South Africa

Dear Dr. Cuadros,

Thank you for submitting your manuscript to PLOS Global Public Health. After careful consideration, we feel that it has merit but does not fully meet PLOS Global Public Health’s publication criteria as it currently stands. Therefore, we invite you to submit a revised version of the manuscript that addresses the points raised during the review process.

We look forward to receiving your revised manuscript.

Kind regards,

Sanghyuk Shin

Academic Editor

Journal Requirements:

Additional Editor Comments (if provided):

I have reviewed the revised manuscript and response to the reviewer comments, and all of the major issues identified during the initial review have been addressed. I applaud the authors for a thorough and thoughtful response to the prior review. Two minor issues remain:

1. Following reporting guidelines for molecular epidemiology studies, the title should clearly state that this is a molecular epidemiology study. Please see:

Field N, Cohen T, Struelens MJ, Palm D, Cookson B, Glynn JR, et al. Strengthening the Reporting of Molecular Epidemiology for Infectious Diseases (STROME-ID): an extension of the STROBE statement. The Lancet Infectious Diseases. 2014 Apr;14(4):341–52.

2. The last sentence in the abstract appears to be a run-on sentence, and it contains assertions about the effect of an intervention that is not directly part of this study. Given that study is observational and descriptive in nature, please rephrase this sentence to be more nuanced about the impact of a hypothetical intervention. The last sentence of the Conclusion (lines 427 - 430) should likewise be revised, since the intervention mentioned in that sentence was not tested in this study.
---

## [Editor Report · Decision Letter 2]

15 Nov 2021

The role of high-risk geographies in the perpetuation of the HIV epidemic in rural South Africa: A spatial molecular epidemiology study

PGPH-D-21-00177R2

Dear Dr. Cuadros,

We're pleased to inform you that your manuscript has been judged scientifically suitable for publication and will be formally accepted for publication once it meets all outstanding technical requirements.

Within one week, you'll receive an e-mail detailing the required amendments. When these have been addressed, you'll receive a formal acceptance letter and your manuscript will be scheduled for publication.

An invoice for payment will follow shortly after the formal acceptance. To ensure an efficient process, please log into Editorial Manager at https://www.editorialmanager.com/pgph/ click the 'Update My Information' link at the top of the page, and double check that your user information is up-to-date. If you have any billing related questions, please contact our Author Billing department directly at authorbilling@plos.org.

Kind regards,

Sanghyuk Shin

Academic Editor
